# Recent Advances of Cellulase Immobilization onto Magnetic Nanoparticles: An Update Review

**Kamyar Khoshnevisan** [1,*], **Elahe Poorakbar** [2], **Hadi Baharifar** [3] **and Mohammad Barkhi** [4]

1   Biosensor Research Center, Endocrinology and Metabolism Molecular-Cellular Sciences Institute, Tehran University of Medical Sciences, Tehran 1411713137, Iran
2   Department of Biology, Faculty of Sciences, University of Payame Noor, Tehran 19395-3697, Iran; epoor2000@yahoo.com
3   Department of Medical Nanotechnology, Applied Biophotonics Research Center, Science and Research Branch, Islamic Azad University, Tehran 1477893855, Iran; baharifar.h@gmail.com
4   Zar Center, University of Applied Science and Technology (UAST), Karaj 1599665111, Iran; mbarkhi@gmail.com
*   Correspondence: k-khoshnevisan@razi.tums.ac.ir or kamyar.khoshnevisan@gmail.com; Tel.: +964-9888220068; Fax: +964-9888220052

**Abstract:** Cellulosic enzymes, including cellulase, play an important role in biotechnological processes in the fields of food, cosmetics, detergents, pulp, paper, and related industries. Low thermal and storage stability of cellulase, presence of impurities, enzyme leakage, and reusability pose great challenges in all these processes. These challenges can be overcome via enzyme immobilization methods. In recent years, cellulase immobilization onto nanomaterials became the focus of research attention owing to the surface features of these materials. However, the application of these nanomaterials is limited due to the efficacy of their recovery process. The application of magnetic nanoparticles (MNPs) was suggested as a solution to this problem since they can be easily removed from the reaction mixture by applying an external magnet. Recently, MNPs were extensively employed for enzyme immobilization owing to their low toxicity and various practical advantages. In the present review, recent advances in cellulase immobilization onto functionalized MNPs is summarized. Finally, we discuss enhanced enzyme reusability, activity, and stability, as well as improved enzyme recovery. Enzyme immobilization techniques offer promising potential for industrial applications.

**Keywords:** cellulase immobilization; magnetic nanoparticles; stability; functionalized nanoparticles

## 1. Introduction

The environmental pollution produced by fossil fuels, the increasing growth of population, and the expensive costs of traditional energy sources compel researchers to develop novel approaches toward ecofriendly and biodegradable energy sources. Biomass, specifically cellulose, nature's most abundant biopolymer, is a low-cost energy source which can be degraded as biomaterials to yield chemical products applicable in many industrial applications [1,2].

Cellulosic enzymes such as cellulases are catalysts which convert cellulose to glucose, and are widely used in different industries, including food, pulp and paper, laundry, beverages, textile, agriculture, pharmaceutics, medicine, and especially in biofuel production [3]. Glucose is the main product of cellulose conversion, which is applied as a precursor for the production of various valuable products. Cellulase, which is synthesized by microorganisms including bacteria and fungi [4,5], is the most powerful hydrolyzing enzyme and can be easily employed [6].

Chemical, physical, and biological methods were employed for cellulosic hydrolysis, from which the enzymatic conversion gained much attention because of its mild reaction conditions. Therefore,

these methods provide high yield with no inhibitory by-products and are considered environmentally friendly. Cellulases are highly selective catalysts and the degradation process is naturally carried out in pH 4.5–5.5 at 40–50 °C [7–9].

Cellulases are responsible for biochemical conversion processes and convert the lignocellulosic biomass (hemicellulose and cellulose) into an intermediate sugar, which further acts as the substrate for ethanol production [10–14]. Biocatalysts have limitations such as limited availability, substrate scope, and operational stability [15]. Recent findings can help scientists overcome these limitations. The main challenge in the application of biocatalysts is their high cost; therefore, reusability and recovery of the enzymes are two significant factors that should be considered for industrial applications [16,17]. In industrial processes, enzymatic reactions are generally performed in high-temperature conditions which can lead to changes in the natural structure of cellulase [18,19]. For this reason, enzyme properties must be greatly improved. Immobilization is a powerful tool to increase the stability and reusability of enzymes.

Enzyme immobilization on support materials is a well-established approach for enhancement of enzyme features such as activity, stability, reusability, purification, reduction of inhibition, and selectivity. Enzyme immobilization on a solid support provides good distribution of the catalysts with less aggregation. On the other hand, covalent binding between the support and the enzyme results in increased enzymatic stability, which in turn leads to enhanced enzymatic activity [1,20,21].

Different methods for enzyme immobilization exist, including covalent binding, adsorption, ionic bonding, entrapment, and encapsulation [22–27]. There are also numerous approaches to facilitate enzyme immobilization on nanomaterials, including enzymatic modifications, enzymatic immobilization and biosensor development [28,29], enzymatic degradation, enzyme nanoparticles, and enzyme mimics of nanomaterials [30–33]. Among several available nanoparticles, magnetic nanoparticles (MNPs) received more attention owing to their advantageous features including low toxicity and high surface area, which allows a large number of enzyme molecules to be loaded to their surface [30,34,35].

In recent decades, most research studies on cellulase immobilization illustrated that the enzyme structure was improved, and bound cellulase maintains high activity for a long time. Furthermore, the immobilized cellulase is more resistant to structural alterations induced by increased temperature [36–39]. The activity of bounded cellulase was shown to be higher at most pH values than the free form due to enhanced stability [40–42].

$Fe_3O_4$ MNPs obtained great attention due to their low toxicity, simple preparation, unique size, strong magnetic properties, and proper physical properties, as well as simple recovery from the media with an external magnetic field [43,44]. Although, magnetite nanoparticles tend to agglomerate and be easily oxidized upon air exposure, it is very important to functionalize its surface in order to avoid oxidization.

The enzymes can bind to the surface of MNPs through van der Waals and electrostatic forces, hydrophobic, or π–π stacking connections by means of non-covalent binding. However, the main challenge of non-covalent immobilization is protein leakage from the surface of the MNPs. Thus, the covalent binding using cross-linkers is widely applied to resolve this problem. Among the cross-linkers, glutaraldehyde is commonly employed as the coupling agent for covalent cellulase immobilization to the support since it is soluble in aqueous solvents and provides firm inter- and intra-covalent bonds [45–49].

A summary of different methods used for MNP functionalization for cellulase immobilization is presented in Figure 1. These functionalization methods are further described and their advantageous features are thoroughly discussed. The current review is focused on recent findings on immobilized cellulases on MNP supports with various functionalized groups and their advantages.

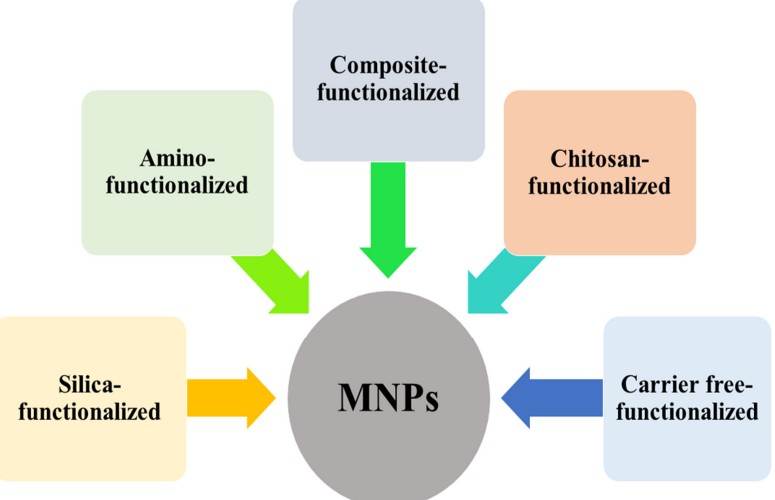

**Figure 1.** An overview of magnetic nanoparticles (MNPs) functionalized by different methods.

## 2. Cellulase Immobilization on Silica-Functionalized MNPs

Silica functionalized MNPs provide suitable supports with low aggregation in solutions. Silica functionalization enhances biocompatibility, as well as thermal and chemical stability, of the MNP surface [34,50]. Co-immobilization of tri-enzymes consisting of cellulase, pectinase, and xylanase was investigated for industrial applications. The results revealed that kinetic parameters (i.e., $V_{max}$ and $K_m$) of the tri-enzyme were not affected by the immobilization process. It can be concluded that the immobilization process significantly enhances thermal and chemical stability [18].

Zhang et al. reported an enzymatic catalysis with chemocatalysis in an iPrOH/water solvent mixture by employing a novel approach for green conversion of ionic liquid (IL) while pretreating cellulose with 5-hydroxymethylfurfural (HMF). This pretreatment of media converted cellulose to glucose and glucose to HMF, while HMF and glucose yields were 43.6% and 86.2%, respectively (Scheme 1). The results obtained from enzymatic cascades and reaction systems prove that this method can be applied as an operative route for biomass energy maintenance [51].

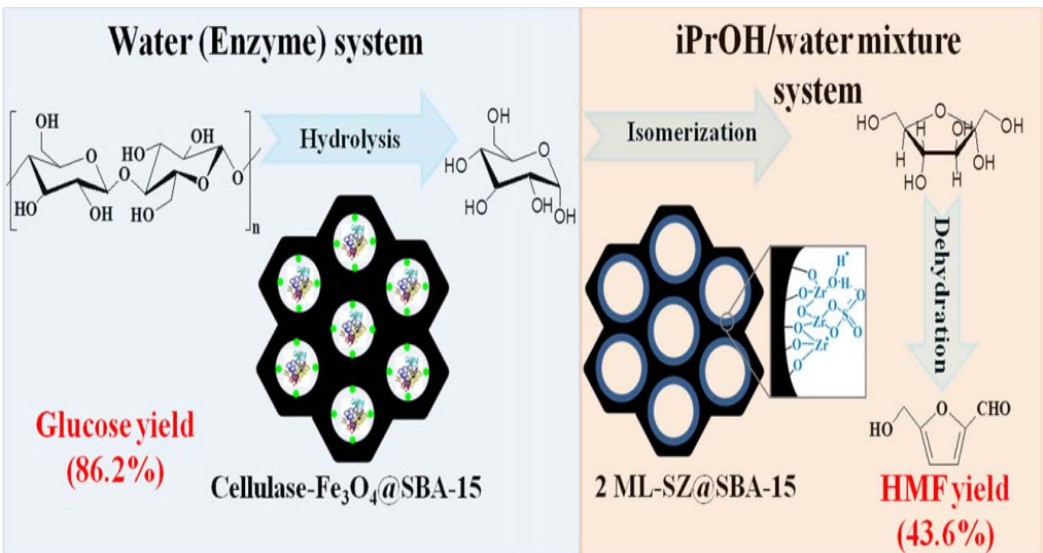

**Scheme 1.** Schematic representation of the cascaded enzymatic and chemical steps for ionic liquid (IL) pretreated cellulose into 5-hydroxymethylfurfural (HMF) in water (enzyme) and iPrOH/water solvent mixtures with enzyme and SBA-15 grafted sulfated zirconium dioxide (SZ) conformed monolayers, respectively. Reproduced with permission from Reference [51].

In another study, *Trichoderma reesei* cellulase was immobilized on two different nanomatrices (MNPs and silica nanoparticles (SNPs)) to improve enzymatic efficiency. Then, 1-ethyl-3-methylimidazoliumacetate [EMIM][Ac] was applied as an IL for cellulase immobilization and was used for pretreatment of sugarcane bagasse and wheat straw. The results obtained from this study revealed a great hydrolysis yield (89%). This process, due to IL reusability and the enhanced stability of the immobilized enzyme, can be potentially used for biorefineries [52].

The immobilization and characterization of holocellulase from *Aspergillus niger* on five different nanoparticles (NPs) via available methods was reported by Kuma et al. Enzyme molecules were covalently immobilized on magnetic enzyme–nanoparticle complexes (MENC), and the results obtained from this study revealed that the immobilization of indigenous enzymes and their consumption can be used for saccharification of paddy straw [53].

Jia et al. established and applied novel MNP cross-linked cellulase aggregates (Figure 2) in order to improve enzyme reusability and stability for biomass bioconversion. The immobilized cellulase further represented suitable activity and stability following reusability in biomass applications [54].

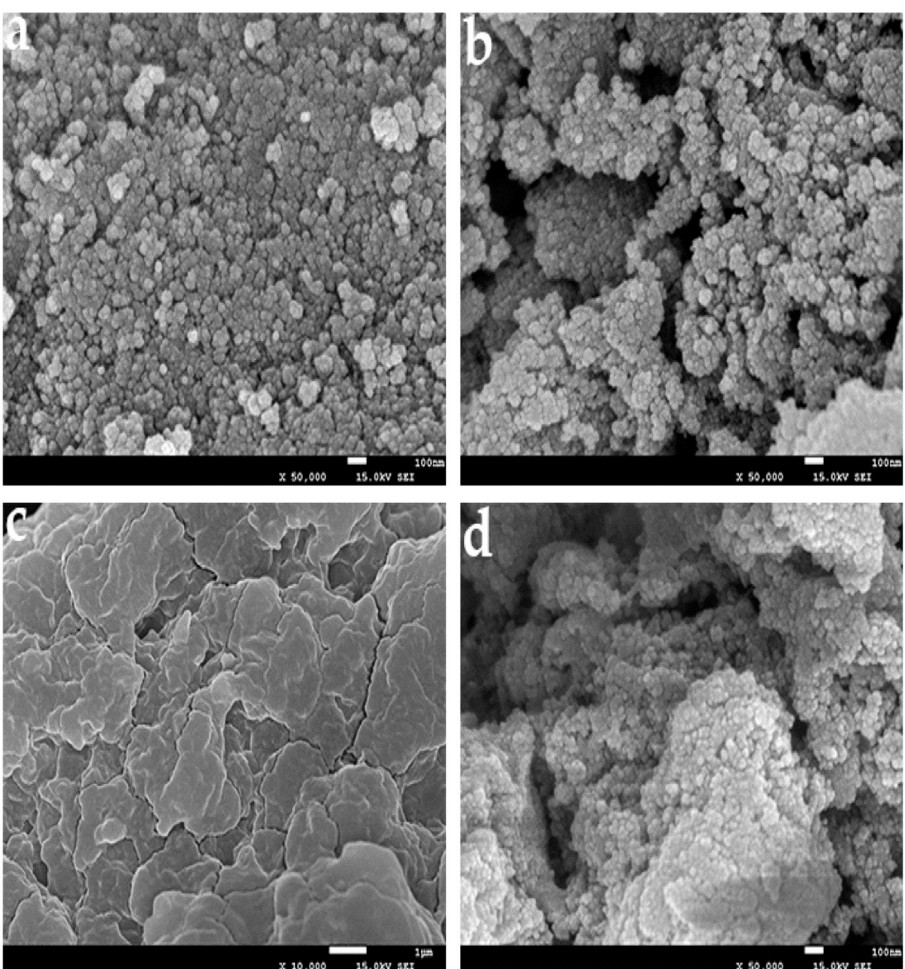

**Figure 2.** Scanning electron microscopy (SEM) images of (**a**) $Fe_3O_4$; (**b**) 3-aminopropyl triethoxysilane (APTES)-$Fe_3O_4$; (**c**) cellulase cross-linked enzyme aggregates (CLEAs); (**d**) magnetic cellulase CLEAs. Reproduced from Reference [54].

The key parameters of enzyme immobilization such as pH, temperature, efficiency, reusability, and coupling agents, as well as supports and substrates, are presented in Table 1. As mentioned before, glutaraldehyde is the most commonly used cross-linker for enzyme immobilization. Our findings suggest that biocompatible, biodegradable, and nontoxic polymers such as polyethylene glycol (PEG) can also be used as an alternative cross-linker for this process.

**Table 1.** Cellulase immobilization on silica-functionalized magnetic nanoparticles (MNPs).

| Enzyme | Substrate | Support | Coupling Agent | Amount * | pH | Temperature | Reusability | | Ref. |
|---|---|---|---|---|---|---|---|---|---|
| | | | | | | | Cycle | Efficiency | |
| Pectinase Xylanase Cellulase | Polygalacturonic acid Xylan CMC | ($Fe_3O_4$) on APTES | Glutaraldehyde | 12, 5, 31 mg/mL | 4.8 | 70 °C | 5 | 87%, 69%, and 58% | [18] |
| Cellulase | Cellulose | $Fe_3O_4$ encapsulated with SBA-15 | PEG-1000 | 1.6 mg | 4.8 | 25–85 °C | 5 | 87.5% | [51] |
| Cellulase | Wheat straw and sugarcane | [EMIM][Ac] functionalized-MNPs and SNP | Glutaraldehyde | 10 and, 7.5 mg/mL | 3.5–9.5 | 20–80 °C | 10 | 85% and 76% | [52] |
| Holocellulase | FP, CMC, and xylan | APTES-$Fe_3O_4$ | Glutaraldehyde | 2 mg/mL | 3–7 | 40–80 °C | 2 | 60–80% | [53] |
| Cellulase | CMC | APTES-$Fe_3O_4$ | Glutaraldehyde | 176 mg/g | 3–8 | 30–80 °C | 6 | 88% | [54] |

Abbreviations: 1-ethyl-3-methylimidazoliumacetate [EMIM][Ac]; silica nanoparticles (SNPs); filter paper (FP); polyethylene glycol (PEG); carboxyl methyl cellulose (CMC); 3-aminopropyltriethoxysilane (APTES); * amount of immobilized enzyme.

## 3. Carrier Free Cellulase Immobilization Strategy

In recent years, MNPs were commonly applied as a solid support for recovery systems [55]. Carrier-free immobilization approaches like co-immobilization by cross-linked enzyme crystals (CLECs) and cross-linking enzyme aggregates (CLEAs) are conceivable methods to boost the enzyme stability [56–59]. Tri-enzyme was co-immobilized on MNPs to improve reusability by cross-linking with glutaraldehyde. In this study, co-immobilized MNPs remained stable for more than a month at 5 °C and also retained activity for up to four cycles. It was suggested that this platform can be effectively applied for the extraction of different plants [19]. In another similar study, enzymes including xylenes, cellulases, and amylases were derived from bacteria and applied for cellulase immobilization on MNPs to enhance stability and facilitate reusability (Figure 3). The results obtained demonstrated that the enzyme-coupled MNPs exhibited excellent stability and recovery. Tri-enzyme immobilized MNPs can be potentially applied in plant biomass production [60].

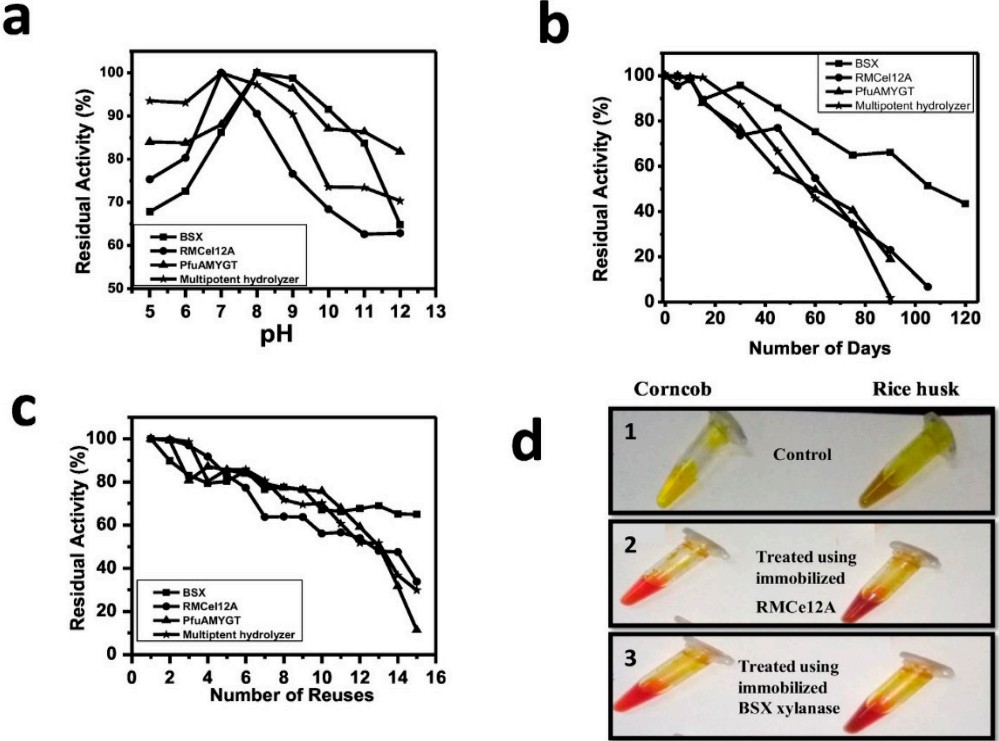

**Figure 3.** Functional characterization of immobilized enzyme-carrying magnetic nanoparticles (MNPs) and untreated (control) nanoparticles where fitting. The text insets describe individual plots or photographs relating to nanoparticles carrying BSX, RMCel12A, or PfuAmyGT, or all the tri-enzymes (multipotent). (**a**) Effect of pH; (**b**) variations in stability upon storage; (**c**) reusability of immobilized enzyme preparations; (**d**) visual evidence for degradation of biomass (corn cob or rice husk) by RMCel12A or BSX. Reproduced with permission from Reference [60].

Different MNPs were prepared and characterized, while the significance of MNP characterization was exclusively discussed by Schwaminger and his colleagues (Scheme 2). Moreover, electrostatic and hydrophobic interactions between the enzyme molecules and MNPs were also investigated. Results from infrared (IR) spectroscopy revealed a high affinity for β-sheet formation in the tertiary structure content of enzyme molecules. In this study, cellulase loading on different MNPs was studied, and the results showed that higher loading efficiencies would be achieved by using a higher α-helical section [61].

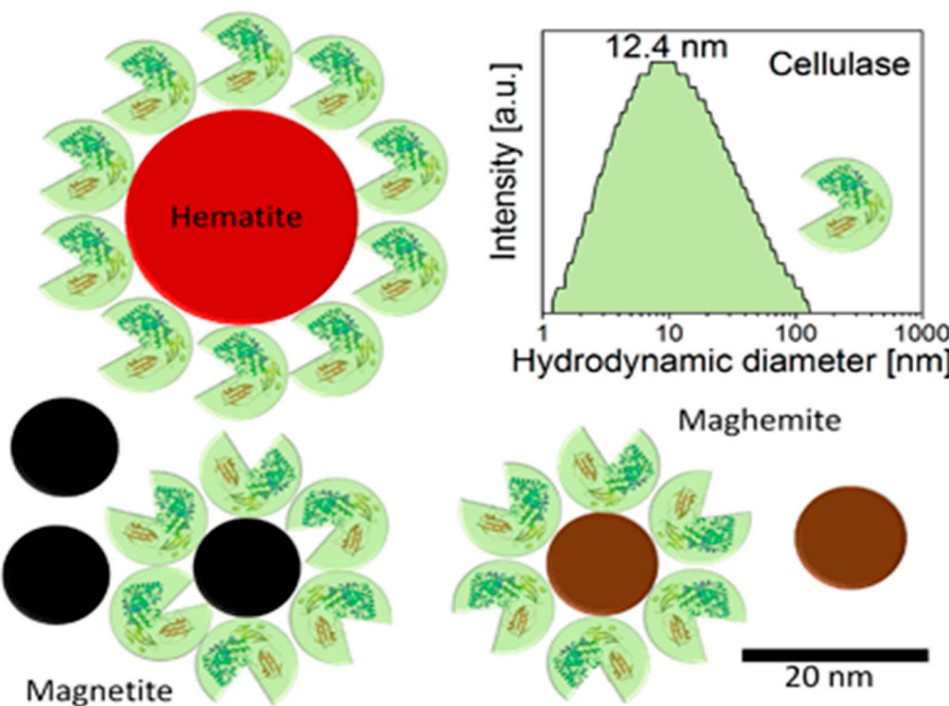

**Scheme 2.** Schematic representation of cellulase immobilization onto three types of MNPs. Reproduced with permission from Reference [61].

In another study, fungus cell filtrate was applied to synthesize MNPs, which was further characterized. The results obtained from this study revealed that the free enzyme was more efficient than the immobilized form and that cellulase molecules retained high activity following immobilization. The authors suggested that cellulase immobilization on MNPs provides good reusability, making the process more efficient for sustainable bioethanol production [62]. As summarized in Table 2, the significant parameters of cellulase immobilization such as pH, temperature, reusability, and coupling agents, as well as supports and substrates, are presented for carrier-free immobilization approach.

**Table 2.** Carrier-free MNP systems for cellulase immobilization.

| Enzyme | Substrate | Support | Coupling Agent | Amount | pH | Temperature | Reusability | | Ref. |
|--------|-----------|---------|----------------|--------|-----|-------------|-------------|------------|------|
| | | | | | | | Cycle | Efficiency | |
| Cellulase, pectinase, and xylanase | CMC, pectin, and xylan | $Fe_3O_4$ | Glutaraldehyde | $5.06 \pm 0.46$ mg/mL $3.39 \pm 0.12$ mg/mL $2.95 \pm 0.14$ mg/mL | 5.5 | 55–75 °C | 4 | $80.25 \pm 1.03\%$ $84.76 \pm 1.71\%$ $75.62 \pm 0.76\%$ | [19] |
| Xylanase, cellulase, amylase | Xylan, CMC, starch | MNPs | Glutaraldehyde | 3 mg/mL | 2–12 | Thermostable up to 70 °C | 13 | 69, 48, and 50% | [60] |
| Cellulase | N/A * | magnetite, maghemite, and hematite MNPs | N/A | $0.6$ g·g$^{-1}$ | N/A | NA | N/A | N/A | [61] |
| Cellulase | Microcrystalline | $Fe_3O_4$ | Glutaraldehyde | 250 mg | N/A | 27 °C, 40 °C, 50 °C and 60 °C | 3 | 52% | [62] |

* Not available.

## 4. Cellulase Immobilization on Amino-Functionalized MNPs

Functionalization of MNPs is a commonly used strategy for tri-enzyme immobilization via covalent bonding. Amines are among the most common functionalized groups that can be linked to proteins via cross-linking agents. Cellulase (from *Trichoderma reesei*) and pectinase (from *Aspergillus aculeatus*) were simultaneously immobilized on amino-functionalized MNPs (AMNPs) for antioxidant extraction from waste fruit peels. This immobilization method led to increased thermal stability, half-life, and $V_{max}$ for both enzymes; however, it caused a slight decrease in their activity. Immobilized enzymes on MNPs show increased reusability of the biocatalyst. Results showed that glutaraldehyde's concertation is an important factor affecting the activity of immobilized enzyme [63]. In a similar study, pectinase and cellulase were immobilized on AMNPs and utilized in the extraction of tomato peel lycopene. The immobilization process decreased enzyme activity while increasing its stability. Ultrasonic irradiation is used to highly activate immobilized cellulase, as well as increase the efficiency of biocatalyst. Results also showed that the biocatalyst decreased extraction time in comparison to free enzyme forms [64]. Hyperactivity of immobilized cellulase on AMNPs was investigated in another study. Application of ultrasound irradiation enhanced cellulase activity up to 3.6-fold. Sonication also increased $V_{max}$ and decreased $K_m$ of cellulase. The results obtained from this study showed that MNP/enzyme ratio, concentration of cross-linking agent, and cross-linking time affected the enzyme activity [65].

Co-immobilization of cellulase and lysozyme on AMNPs was performed for extraction of lipids from microalgae (Scheme 3). Their findings showed that enzyme stability and catalytic efficiency were increased; however, the activity and kinetic parameters of both enzymes decreased following immobilization [66].

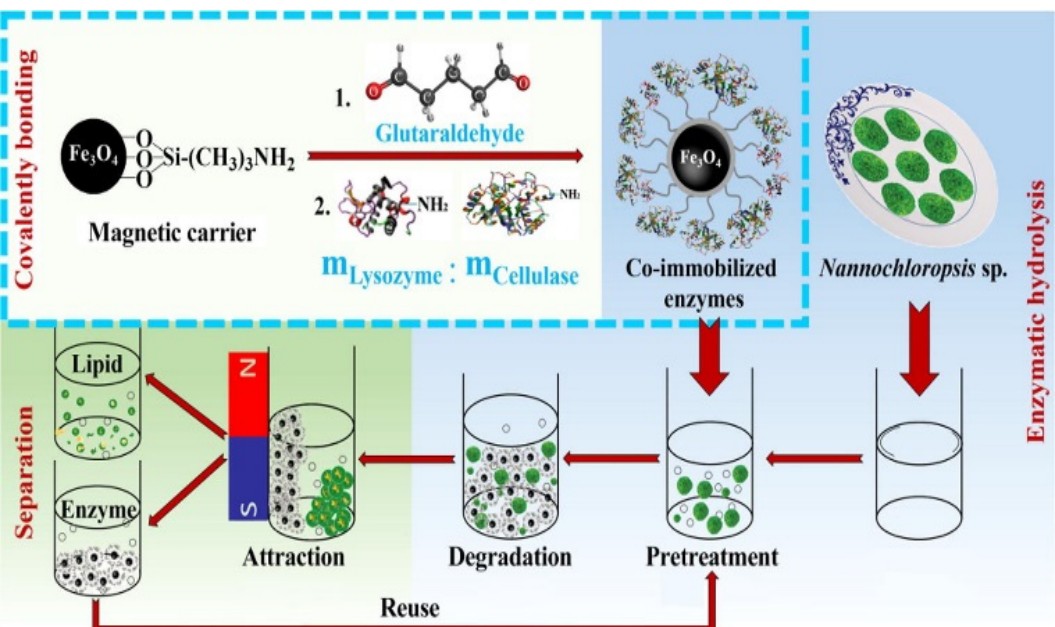

**Scheme 3.** Co-immobilization of cellulase and lysozyme on amino-functionalized MNPs (AMNPs) for extraction of lipid. Reproduced with permission from Reference [66].

It is believed that co-immobilization of cellulose is greatly influenced by the enzyme concentration on the MNP surface. In addition, the immobilization process increases the efficiency of biocatalyst via multiple enzymes; however, applicable enzyme concertation is limited and enzyme activity is reduced due to low availability of surface area. As summarized in Table 3, enzyme immobilization on AMNPs via covalent bonding can increase thermal and chemical stability while generally reducing enzyme activity. MNPs provide biocatalyst reusability, while sonication hyperactivates immobilized cellulase which can compensate for the enzyme activity. It can be concluded that enzyme immobilization on AMNPs can be used for industrial applications.

**Table 3.** Cellulase immobilization on amino-functionalized MNPs (AMNPs).

| Enzyme | Substrate | Support | Coupling Agent | Amount | pH | Temperature | Reusability | | Ref. |
|---|---|---|---|---|---|---|---|---|---|
| | | | | | | | Cycle | Efficiency | |
| Pectinase, cellulase | Carboxymethyl cellulose (CMC) | AMNPs | Glutaraldehyde | 9 and 3 mg/mL | 6.5 | 50–70 °C | 8 | 87% and 82% | [63] |
| Pectinase, cellulase | pectin, cellulose | AMNPs | Glutaraldehyde | 50 mg | 5 | 25–35 °C | 8 | 85% and 80% | [64] |
| Cellulase | cellulose | AMNPs | Glutaraldehyde | N/A | 3–8 | 30–80 °C | 7 | 58% | [65] |
| Cellulase, lysozyme | cell walls | AMNPs | Glutaraldehyde | 0.5 mg | 3–7 | 60–80 °C | 6 | 78.1% and 69.6% | [66] |
| Cellulase | CMC | Cu/AMNPs | APTES | N/A | 2–7 | 20–80 °C | 5 | 73% | [67] |

Amino-functionalized MNPs (AMNPs); 3-aminopropyl-triethoxysilane (APTES).

In a recent study, AMNPs in combination with copper (Cu) (as an affinity ligand) were employed for immobilization of *A. niger*-derived cellulase. Metal affinity ligands are commonly used due to their high chemical stability, low cost, and modifying capability. Loading concentration and cellulase activity were investigated by full factorial design, considering pH, and Cu/MNP and enzyme/MNP ratios as independent variables. Obtained data revealed that Cu improved the immobilized enzyme's activity, storage stability, and loading capacity (i.e., 164 mg/g MNPs). The biocatalyst remained stable under a wide range of pHs and temperatures [67].

## 5. Cellulase Immobilization on Composite-Functionalized MNPs

MNPs used as enzyme immobilization supports can be coated with various nanomaterials. MNP coating prevents nanoparticles oxidation, improves enzyme immobilization efficacy, and may decrease toxicity of the support materials. The coating process should not alter the magnetic properties of nanoparticles, which are essential for the biocatalyst's reusability through a magnetic field. Different composite coatings were prepared for cellulase immobilization, and the effects of coating on enzyme activity and stability were studied.

In recent years, metallic, metallic oxide, and carbon-related materials were widely applied and blended with different polymers to prepare novel nanocomposites for enzyme immobilization. Specifically, gold, MgO, and graphene oxide (GO) were blended with different polymers such as PEG, glutamic acid, and poly(methyl methacrylate) for enhancing the enzyme stability.

Immobilized cellulase on MNPs coated with layered double hydroxide (LDH) nanosheets were used to reduce the magneto-induced effect on the enzyme. Results showed that utilizing nanocomposite materials could increase specific enzyme activity and loading efficiency, but could reduce enzyme reusability. Immobilized cellulase also demonstrated higher stability in a wide range of temperatures and pH values [68]. MgO-coated MNPs were used as a support for covalent immobilization of cellulase from *Chlorella* sp. CYB2. MgO played a significant role for improvement of immobilization yield, activity recovery, and hydrolysis of the substrate [69].

In another study, poly(methyl methacrylate)-coated MNPs were used for cellulase immobilization as shown in Scheme 4. Enzyme stability and activity were affected by the immobilization process as detailed in Table 4. The results obtained revealed that poly(methyl methacrylate) as a coating polymer did not have any significant effect on the particles' magnetic properties (Figure 4) [70].

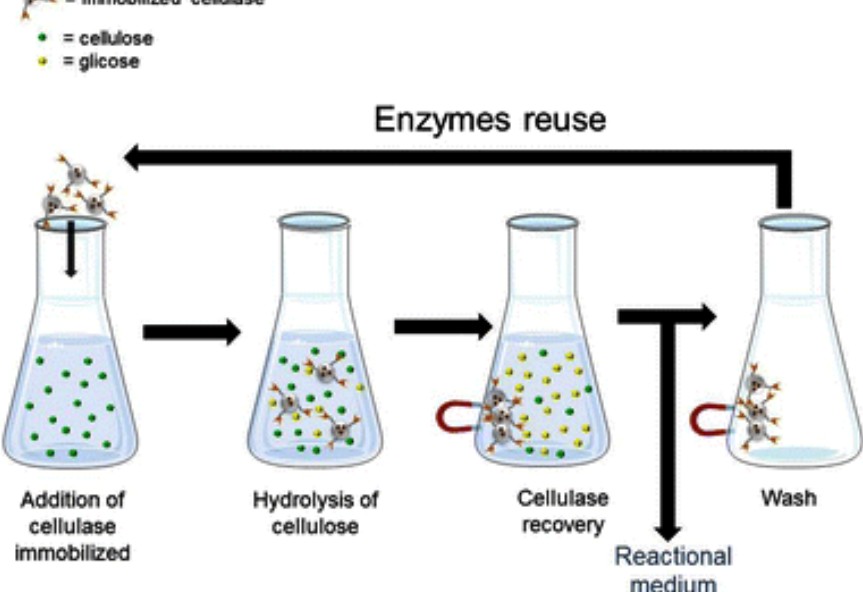

**Scheme 4.** Overview of hydrolysis of cellulose by poly(methyl methacrylate)-coated MNPs applied for enzyme immobilization and enzyme reusability. Reproduced with permission from Reference [70].

**Table 4.** Cellulase immobilization on composite functionalized MNPs.

| Enzyme | Substrate | Support | Coupling Agent | Amount | pH | Temperature | Reusability | | Ref. |
|--------|-----------|---------|----------------|--------|-----|-------------|-------|------------|------|
| | | | | | | | Cycle | Efficiency | |
| Cellulase | CMC | $(Fe_3O_4)$ layered double hydroxides (LDHs) | Glutaraldehyde | 1.2 g/L | 5.5 | 50 °C | 6 | 31.8% | [68] |
| Cellulase | cellulose | $MgO–Fe_3O_4$ | Xylan aldehyde | 150 mg/g | 4.5–6.5 | 50–70 °C | 7 | 84.5% | [69] |
| Cellulase | CMC | Poly(methyl methacrylate) MNPs | N/A | 5% (*w/v*) | 3–8 | 35–75 °C | 8 | 69% | [70] |
| Cellulase | Microcrystalline cellulose or filter paper | $Fe_3O_4$-$NH_2$@4-arm-PEG-$NH_2$ | Glutaraldehyde | 132 mg/g | 3–7 | 30–80 °C | 6 | 76%. | [45] |
| Cellulase | Microcrystalline cellulose or filter paper | GO@$Fe_3O_4$@4arm PEG $NH_2$ | Glutaraldehyde | 2–8 mg | 3.5–5.5 | 30–80 °C | 7 | 65% and 70% | [71] |
| Cellulase | Microcrystalline or filter paper | Glu@PEGylated mAu@PSN | Glutamic acid | 25 mg | 3–8 | 35–75 °C | 5 | 76% | [72] |

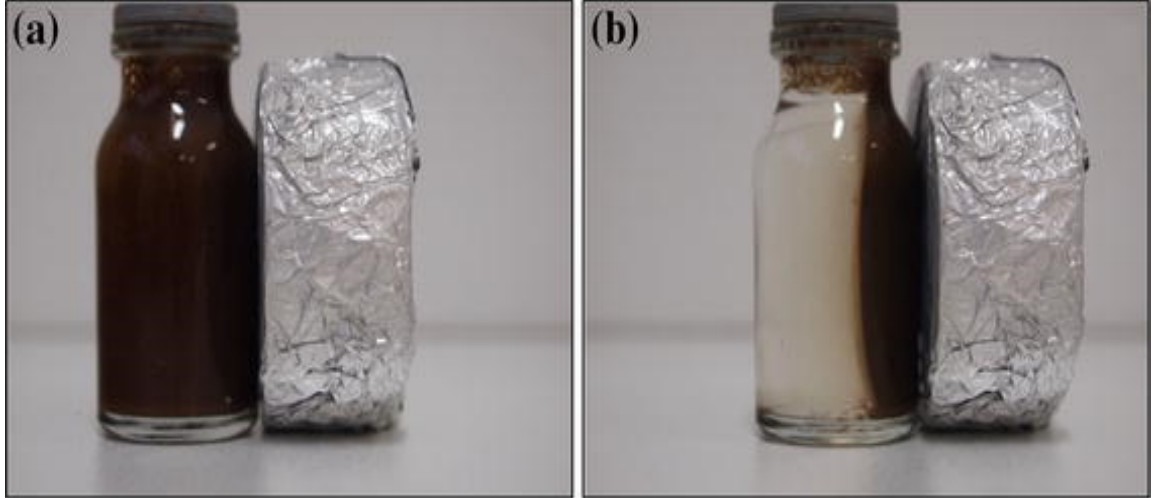

**Figure 4.** Magnetic behavior of poly(methyl methacrylate)-coated MNPs before (**a**) and after (**b**) separation. Reproduced with permission from Reference [70].

Four-arm dendritic polymers composed of PEG-NH$_2$ (Scheme 5) were used as coating materials for immobilization of cellulase derived from *Trichoderma viride* onto MNPs. The enzyme was covalently bonded to the dendrimer via coupling agents. The dendrimer improved the thermal stability and activity of cellulase [45]. GO-decorated four-arm PEG-NH$_2$ was applied as a coating composite of MNPs in another study (Scheme 6). The obtained results showed that polymers with higher molecular weights can increase loading capacity, enzymatic activity, and storage stability of cellulase [71].

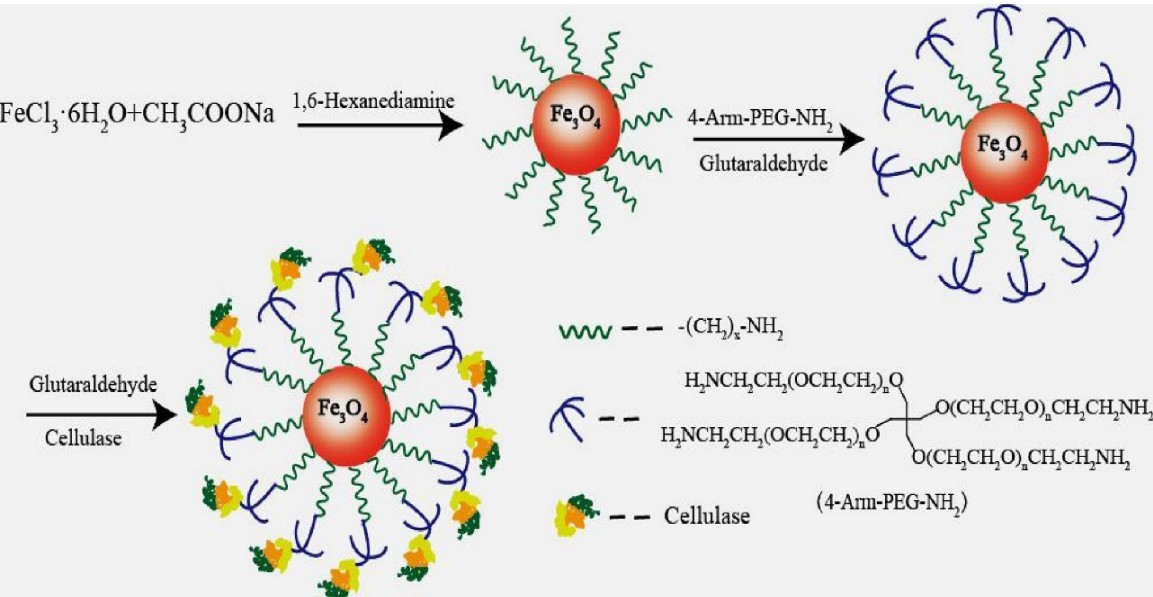

**Scheme 5.** Preparation process of four-arm dendritic polymers comprising polyethylene glycol (PEG)-NH$_2$. Reproduced with permission from Reference [45].

Core–shell magnetic gold mesoporous silica was exploited as a support for cellulase immobilization (Scheme 7). Thermal and chemical stabilities were considerably augmented in a wide range of pH values and temperatures. Vibrating-sample magnetometer (VSM) study results showed that the magnetic behavior of MNPs was not altered following the coating process [72].

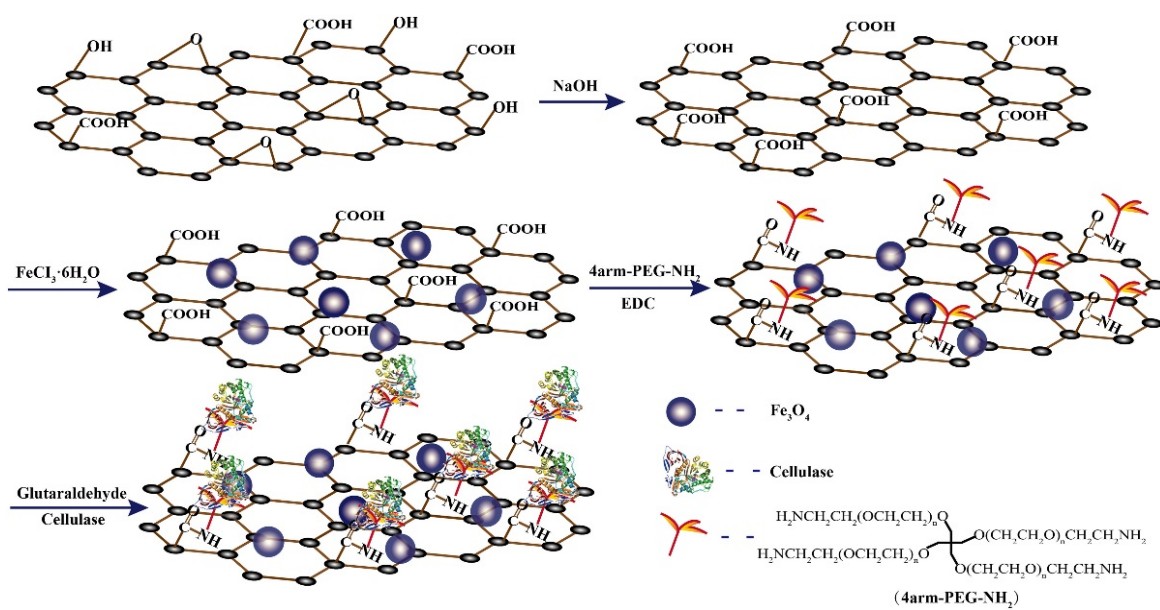

**Scheme 6.** Graphene oxide (GO)-decorated four-arm PEG-NH$_2$ coating composite of MNPs. Reproduced with permission from Reference [71].

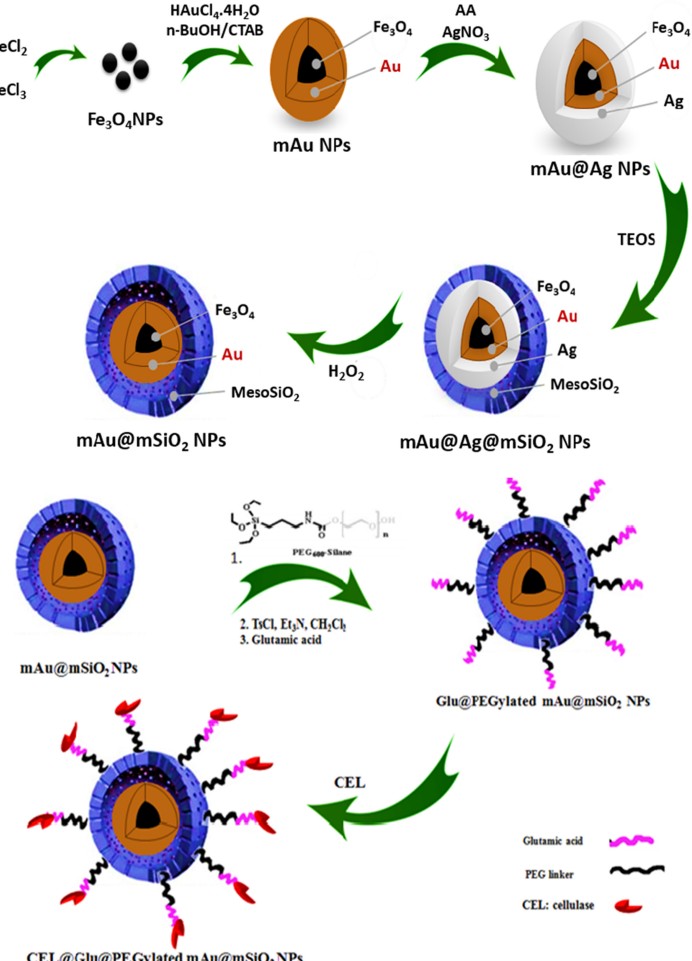

**Scheme 7.** Cellulase immobilization on core–shell magnetic gold mesoporous silica support. Reproduced with permission from Reference [72].

As presented in Table 4, cellulase was immobilized on MNP surfaces via different coating materials. Polymers are extensively applied as preferred coating materials for cellulase immobilization. The immobilization of cellulase on MNPs via covalent binding improves biocatalyst activity and stability, regardless of coating type.

## 6. Cellulase Immobilization on Chitosan-Functionalized MNPs

Chitosan, a biocompatible and biodegradable polymer, is exclusively used as a coating agent in enzyme immobilization processes. Chitosan can provide a positively charged coating, while acting as a toxicity reducing agent or adhesive enhancer for simple immobilization processes or in vivo applications.

Cellulase enzyme was covalently immobilized on chitosan-coated MNPs (Ch-MNPs) using coupling agents. The immobilization of cellulase decreased enzyme activity compared to free enzyme, while thermal stability and reusability of the enzyme was improved. The immobilization process significantly increased the $K_m$ value and the biocatalyst efficiently hydrolyzed lignocellulosic materials from *Agave atrovirens* leaves with acceptable yield [37]. In a similar study, cellulase was immobilized on MNPs using a cross-linking agent (Scheme 8), and the optimal enzyme loading efficiency and standard recovery ratio were studied [46].

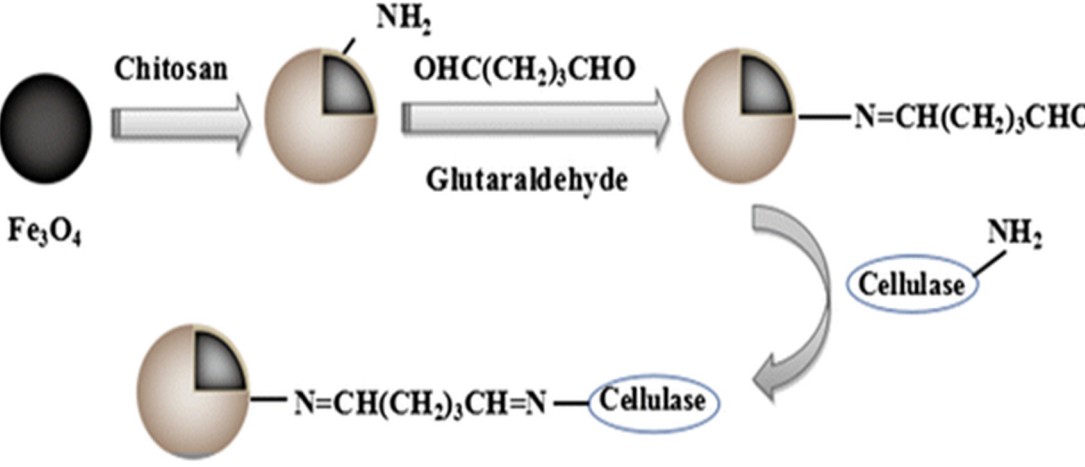

**Scheme 8.** Immobilization of cellulase on chitosan-coated (Ch)-MNPs using glutaraldehyde as a cross-linking agent. Reproduced with permission from Reference [46].

Maximizing the rate of cellulase and xylanase immobilization onto Ch-MNP was studied, and the obtained data showed that 12 mg of protein was cross-linked per gram of MNPs. The results also showed that size distribution, shape, and surface chemistry of MNPs affected the coating process and immobilization efficiency [73]. In the case of enzymatic saccharification, laccase from *Trametes versicolor* was immobilized onto Ch-MNPs (Scheme 9). It was concluded that the catalytic activity, thermal and chemical stability, and the $K_m$ value were improved significantly following enzyme immobilization [74].

Chitosan as a coating material for MNPs was utilized for enzyme cross-linking in order to enhance the stability parameters (Table 5). Chitosan-coated MNPs are prepared in a one-step process which can be a simple and cost-effective method for enzyme immobilization.

**Table 5.** Cellulase immobilization on chitosan-functionalized MNPs.

| Enzyme | Substrate | Support | Coupling Agent | Amount | pH | Temperature | Reusability | | Ref. |
|--------|-----------|---------|----------------|--------|-----|-------------|-------|------------|------|
| | | | | | | | Cycle | Efficiency | |
| Cellulase | CMC | Chitosan-coated MNPs (Ch-MNPs) | Glutaraldehyde | 26.06 mg | 2.5–8.5 | 20–70 °C | 15 | 80% | [37] |
| Cellulase | CMC | Magnetic $Fe_3O_4$–chitosan | Glutaraldehyde | 32.29 mg | 3–7 | 30–70 °C | 5 | 80% | [46] |
| Xylanase and cellulase 1:0.5 | N/A | Chitosan-coated magnetite particles | Glutaraldehyde | N/A | N/A | N/A | N/A | N/A | [73] |
| Laccase | Lignin | Chitosan (C)-MNP | Glutaraldehyde | 25 mg | 2–7 | 25–75 °C | 5 | 50% | [74] |

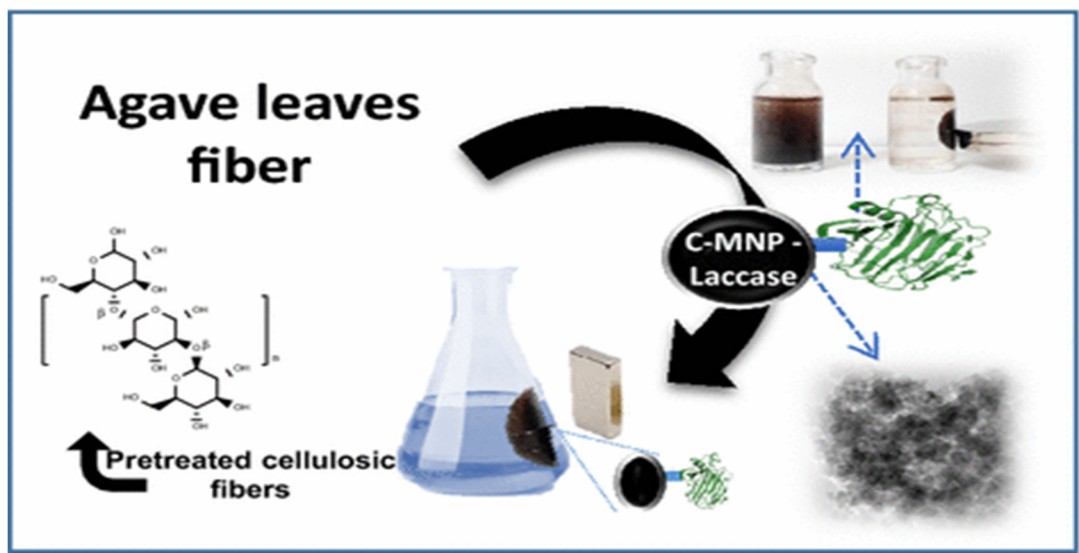

**Scheme 9.** Enzymatic saccharification and reusability of immobilized laccase onto Ch-MNPs. Reproduced with permission from Reference [74].

## 7. An Overview of Principal Factors Affecting Cellulase Immobilization onto MNPs

Among the different methods available for cellulase immobilization on functionalized MNPs, covalent binding by applying glutaraldehyde as the coupling agent is the most efficient process. Different types of functionalized MNPs were studied for cellulase immobilization [39,75–77]. Gold magnetic silica nanoparticles were used as a promising platform for cellulase immobilization. Graphene oxide is another candidate which retains high enzymatic activity. In addition to functionalized groups, chitosan- and silica-coated MNPs could be used as efficient solid supports for biomaterials hydrolysis. The application of ultrasound is another approach for improving the properties of immobilized enzyme. Different functionalization methods of MNPs for the cellulase immobilization approach and the attributes of each approach are outlined in Table 6.

**Table 6.** Recent advances in immobilizing cellulase onto MNPs.

| Different Functionalized MNPs Applied in Cellulase Immobilization Approaches | | |
|---|---|---|
| **Immobilization Approach** | **Attributes** | **Ref.** |
| Silica-based surface functionalization | Enhanced chemical stability while avoiding the aggregation of nanoparticles | [51–54] |
| Composite-based surface functionalization | Providing unique physical and electronic properties and also providing a large surface area for biomolecules to anchor | [69–72] |
| Amino-based surface functionalization | Novel strategies to enhance the enzyme's thermal and chemical stability | [63–67] |
| Chitosan-based surface functionalization | Providing an appropriate surface for biomolecules to anchor | [46,74] |
| Carrier-free immobilization | Novel strategies to improve enzyme activity | [60–62] |

## 8. Summary and Outlook

Biotechnology opened new horizons for human beings, especially in the field of industry. Due to increased environmental pollution, and of the increase in human population, biotechnology employs nano-biomaterials in order to enhance product yield. Cellulase is one of the most widely used biocatalysts that converts cellulosic materials into monosaccharides such as glucose, which are further used for biofuel production. However, biomass conversion to glucose needs to be efficient and

cost-effective; as a result, researchers are focusing on the development of novel approaches with enhanced enzyme reusability and lower cost, along with easy enzyme recovery from the reaction mixture. During the last two years, studies showed that immobilized cellulases on nanocarrier supports provide promising potential as novel nano-biocatalysts which can be further exploited for achieving higher enzyme activity and storage stability in the immobilization processes. Meanwhile, MNPs are suitable nano-carriers to separate enzymes from the reaction mixture by using an external magnet while being efficient by reducing recycling costs. Various MNPs with different functional groups as solid supports exist, including inorganic metals, graphene, chitosan, and organic compounds. Despite recent advances, novel approaches are still required for achieving more efficient nano-biocatalysts. The increasing demand for low-cost immobilized cellulase and the ever-increasing applications in different industries are the main reasons for research attention in this field.

**Author Contributions:** Conceptualization, methodology, writing, original draft preparation, table preparation, review, and editing, K.K.; writing, original draft preparation, table preparation, and review E.P.; writing, original draft preparation, and table preparation, H.B.; original draft preparation, schemes and figures preparation, M.B.

**Funding:** Private funds were applied to carry out this study.

**Acknowledgments:** The authors would like to express special thanks to F. Vakhshiteh, A.K. Bordbar, and H. Stamatis for their practical and valuable collaboration in our previous study.

**Conflicts of Interest:** The authors declare no conflicts of interest.

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
