# Peer review of "Recent Advances of Cellulase Immobilization onto Magnetic Nanoparticles: An Update Review"

_magnetochemistry, doi:10.3390/magnetochemistry5020036_

Round 1

Reviewer 1 Report

Comments:

This manuscript summarized the methods of immobilization cellulase onto magnetic nanoparticles.  As an updated review, the authors summarized different strategies into tables for easier comparison covering pH, temperature and reusability. However, discussions are not solid and conclusive. Therefore, a publication could be supported if the following concerns are addressed.

1.      The authors focus on glutaraldehyde type of covalent immobilization cellulase on the different type of magnetic particles including silica-, and gold-coated magnetic nanoparticles. However, besides the glutaraldehyde as a crosslinking reagent, there are several chemical routes to bond between nanoparticles and cellulase including adsorption, encapsulation, etc. as mentioned in the introduction. I would suggest the authors introduce other methods as well.

2.      The manuscript gives the discussion of immobilization cellulase on the magnetic nanoparticles. However, the authors blend silica and amine functionalization together. The silica surface itself without any surface functionalization is seldom to covalently immobilize enzyme directly. Silica coating provides a platform for easier surface modification.  I would suggest the authors elaborate on different methods more carefully.

3.      In this manuscript, methods described here mostly are based on the amino-functionalization type of MNPs with different strategies to modify amino group on MNPs surface.  If authors could include other functional group strategies such as carboxylic acid type, this manuscript could become more complete.

4.      For the composite functionalized, please explain or highlight the function of the composite structure either. For example, the advantage of gold composition.

5.      In line 242, the author mentions the coating should not alter the magnetic properties. However, the saturation magnetization will decrease with a heavy coating. Please rephrase the description for the magnetic properties.  

6.      In table 1-5, the authors provide a detailed comparison of different studies. However, some parameters are hard to compare. For example, the amount used in different references are hard to compare as they are in different unit. For the enzyme research, it is important to consider the amount required in the application. For the pH range and temperature, the authors could provide a summary of the update studies to improve the activity or reusability. For the overlook, the authors could discuss the advantages of different strategies for different applications with a summary from the references/table to provide readers more takeaway information from this review.

Editorial comment:

1.     References:  In Ref 38, 51, 54, 69, please include page number

Author Response

This manuscript summarized the methods of immobilization cellulase onto magnetic nanoparticles.  As an updated review, the authors summarized different strategies into tables for easier comparison covering pH, temperature and reusability. However, discussions are not solid and conclusive. Therefore, a publication could be supported if the following concerns are addressed.

1.      The authors focus on glutaraldehyde type of covalent immobilization cellulase on the different type of magnetic particles including silica-, and gold-coated magnetic nanoparticles. However, besides the glutaraldehyde as a crosslinking reagent, there are several chemical routes to bond between nanoparticles and cellulase including adsorption, encapsulation, etc. as mentioned in the introduction. I would suggest the authors introduce other methods as well.

The following sentences were added to comply with the comments.

The enzymes can bind to the surface of MNPs through van der Walls and electrostatic forces, hydrophobic or π-π stacking connections by means of non-covalent binding. However, the main challenge of non-covalent immobilization is protein leakage from the surface of the MNPs. Thus, the covalent binding by using cross linkers has been widely applied to resolve this problem. Among the cross linkers,

2.      The manuscript gives the discussion of immobilization cellulase on the magnetic nanoparticles. However, the authors blend silica and amine functionalization together. The silica surface itself without any surface functionalization is seldom to covalently immobilize enzyme directly. Silica coating provides a platform for easier surface modification.  I would suggest the authors elaborate on different methods more carefully.

The methods was carefully checked and corrected the item to comply with the comments.

3.      In this manuscript, methods described here mostly are based on the amino-functionalization type of MNPs with different strategies to modify amino group on MNPs surface.  If authors could include other functional group strategies such as carboxylic acid type, this manuscript could become more complete.

Thank you so much for your consideration. Actually, based on our search on Scopus and ISI, we did not find any paper in this regard.

4.      For the composite functionalized, please explain or highlight the function of the composite structure either. For example, the advantage of gold composition.

The following sentences were added to comply with the comment.

In the recent years, metallic, metallic oxide and carbon related-materials have been widely applied and were blended with different polymers to prepare novel nanocomposites for enzyme immobilization. Herein, gold and MgO and also graphene oxide (GO) were blended to different polymers like PEG for enhancing the enzyme stability.

5.      In line 242, the author mentions the coating should not alter the magnetic properties. However, the saturation magnetization will decrease with a heavy coating. Please rephrase the description for the magnetic properties.

 The description were amended to comply with the comment.

6.      In table 1-5, the authors provide a detailed comparison of different studies. However, some parameters are hard to compare. For example, the amount used in different references are hard to compare as they are in different unit. For the enzyme research, it is important to consider the amount required in the application. For the pH range and temperature, the authors could provide a summary of the update studies to improve the activity or reusability. For the overlook, the authors could discuss the advantages of different strategies for different applications with a summary from the references/table to provide readers more takeaway information from this review.

The unit of amount of immobilized enzyme was corrected to comply with the comment.

The summary table entitled “Summary of recent advances in immobilized cellulase onto MNPs” was added to section 7 to comply with the comment.

Editorial comment:

1.     References:  In Ref 38, 51, 54, 69, please include page number

The references were amended to comply with the comment.

Reviewer 2 Report

English is so wrong that makes evaluation complex.

Just in abstract:

in the biotechnology processES14 including foods, cosmetics, detergents, pulp and paper, and related industrial process operations”

 “…and lake of enzyme reusability”?

Cellulase as a general activity may be in singular, as enzyme should be in plural.

In general introduction

“Enzyme immobilization, to locate enzymes on support materials, is a well-established 74 technology that enhances enzyme features such as activity, stability, reusability, and selectivity.” Explain a little more, this is a review, and add references, reviews if available. They can add also purification of the enzyme and rediction of inhibition.

Explain why using cellulases nanomaterials are preferred (see review from Garcia-Galan)

I am sorry but my English is not good enough to correct the paper. When English and style have been corrected, the paper shopuld be reevaluated, as now the evaluaiton has been just in the first paragraphs.

Author Response

Comments and Suggestions for Authors

 “ in the biotechnology processES14 including foods, cosmetics, detergents, pulp and paper, and related industrial process operations”

  “…and lake of enzyme reusability”?

The sentences were corrected to comply with the comment.

Cellulase as a general activity may be in singular, as enzyme should be in plural.

 “Cellulases” is generally applied for Pectinase, Xylanase and Cellulase.

In general introduction

“Enzyme immobilization, to locate enzymes on support materials, is a well established 74 technology that enhances enzyme features such as activity, stability, reusability, and selectivity.” Explain a little more, this is a review, and add references, reviews if available. They can add also purification of the enzyme and reduction of inhibition.

The sentence was revised based on the comment. 

The manuscript was thoroughly revised and has remarkably improved to comply with the comments.

Magnetochemistry EISSN 2312-7481 Published by MDPI AG, Basel, Switzerland RSS E-Mail Table of Contents Alert
Back to Top